# Homogeneous Polymerization of Kraft Lignin Using an Alkaliphilic Multi-Copper Oxidase (Bilirubin Oxidase) in a Borate Buffer

**DOI:** 10.3390/polym17060779

**Published:** 2025-03-14

**Authors:** Lou Delugeau, Aurèle Camy, Léna Alembik, Philippe Poulin, Sébastien Gounel, Nicolas Mano, Frédéric Peruch, Stéphane Grelier

**Affiliations:** 1Laboratoire de Chimie des Polymères Organiques (LCPO), University Bordeaux, CNRS, Bordeaux INP, UMR 5629, F-33600 Pessac, France; ldelugeau@bordeaux-inp.fr (L.D.); aurele.camy@bordeaux-inp.fr (A.C.); lena.alembik@bordeaux-inp.fr (L.A.); 2Centre De Recherche Paul Pascal (CRPP), UMR CNRS 5301, University Bordeaux, F-33600 Pessac, France; philippe.poulin@u-bordeaux.fr (P.P.); sebastien.gounel@crpp.cnrs.fr (S.G.); nicolas.mano@crpp.cnrs.fr (N.M.)

**Keywords:** kraft lignin, bilirubin oxidase, alkaliphilic medium, coupling

## Abstract

Enzymatic modification of Kraft lignin under alkaline conditions was investigated using bilirubin oxidase (BOD) in borate buffer (pH 10). Control solubilization without enzyme addition revealed a notable increase in molar mass (up to 1.7-fold) and potential borate complexation with lignin hydroxyl groups, as evidenced by thermogravimetric and ^11^B NMR analyses. BOD treatments induced substantial polymerization, with molar mass increases of up to 4-fold for insoluble fractions after 24 h, while soluble fractions exhibited progressive increases over 5 days. Quantitative ^31^P NMR showed reductions in aliphatic and phenolic hydroxyl groups by 20%, suggesting oxidative coupling reactions, particularly through 4-O-5′ and 5-5′ linkages. Solid-state ^13^C NMR confirmed structural changes associated with polymerization. Dynamic light scattering (DLS) indicated the presence of colloidal aggregates, potentially explaining challenges in HSQC NMR signal acquisition. These findings highlight the efficacy of bilirubin oxidase in catalyzing lignin polymerization and underscore the structural impact of borate–lignin interactions in alkaline media, paving the way for advanced lignin valorization strategies.

## 1. Introduction

The biopolymer lignin exhibits the highest content of aromatic subunits on Earth, inducing stiffness to plants. In vitro lignin polymerization is typically catalyzed by laccases or peroxidases, enzymes that trigger oxidative coupling reactions through highly reactive radical intermediates derived from monolignols [1]. During industrial processing, native lignin is extracted and converted into technical lignins, such as Kraft or alkaline lignins, through various methods [2]. To this day, Kraft lignin’s primary use is as an energy source to produce heat and electricity for the pulp industry infrastructures, with the Kraft process accounting for approximately 85% of the chemical extraction methods used today [3,4].

Laccases (*EC 1.10.3.2*) are multicopper oxidases that catalyze the abstraction of one electron from the phenolic hydroxyl group, using molecular oxygen as the terminal electron acceptor, which is reduced to water in the process. The ambivalent nature of laccase activity on lignin structure has been the subject of extensive research. Lignin, being one of the natural substrates of laccases, is oxidized by these enzymes. Munk et al. have extensively reviewed the literature and reported that the resulting oxidized species can lead to various outcomes, including polymerization, chemical modifications (such as oxidation or demethylation), or depolymerization [5].

Currently, it is widely accepted that native laccases alone catalyze the polymerization of lignin. One can mention the study of Mattinen et al. that explored enzymatic polymerization of various lignins by laccase from *Trametes hirsuta* [6]. More recent studies have expanded this knowledge, such as Agustin et al. who evaluated key factors influencing laccase-catalyzed lignin polymerization and grafting reactions, offering valuable insights into optimizing these processes [7]. More recently, Lu et al. proposed a green method for polyphenol polymerization, using tannin-rich extracts as building blocks to promote lignin polymerization through laccase-catalyzed oxidation [8]. Enzymatic lignin modification has gained significant interest due to its ability to operate under mild aqueous conditions, avoiding the use of high pressure, extreme temperatures, or hazardous chemical reagents. This approach enables selective oxidation and depolymerization while preserving key functional groups, making it a more sustainable and controlled alternative to conventional chemical or physical treatments.

Laccases have also been shown to catalyze bond cleavage in low-molar mass phenolic lignin model compounds. However, to enable inter-unit bond cleavage in lignin substrates, a mediator system is required. The enzyme’s redox potential plays a key role in this process—higher redox potentials allow laccases to oxidize a broader range of substrates.

Despite these advances, technical lignins, such as Kraft lignins, often exhibit unpredictable solubility in common solvents, which poses a challenge for enzymatic treatments. There is an inherent incompatibility between the pH range in which technical lignins are soluble (alkaline pH) and the pH range where laccases are typically active and stable (acidic-to-neutral pH). As a result, the enzymatic treatment of lignins using native laccases is not optimal. To address this issue, some research groups have opted to use lignosulfonates, which are water-soluble, as substrates. However, the presence of sulfur can be detrimental in various applications and can also act as an impurity that can degrade carbonization. Other studies employed fractionation methods to obtain lignins of narrower molar mass distributions, modified functionalities compared to native lignins, and enhanced solubility in specific solvents, thereby facilitating both analytical procedures and enzymatic treatments [9,10]. Alternatively, some teams have employed solvent mixtures, such as ionic liquids, to facilitate the enzymatic treatment of lignin [11,12,13].

To overcome these limitations, the use of alkaliphilic enzymes has been explored by several research groups. These enzymes are active and stable under alkaline conditions, making them well-suited for the treatment of technical lignins. Our team recently reviewed studies focusing on the enzymatic treatment of lignin substrates by alkaliphilic laccases [14]. These studies revealed an ambivalence in the action of these enzymes, with observations of both polymerization and depolymerization, and sometimes even simultaneous occurrences of these phenomena.

Among the pool of alkaliphilic oxidases, bilirubin oxidase (BOD, EC 1.3.3.5) emerged as a promising candidate for lignin treatment. This multicopper oxidase is naturally stable under neutral to alkaline pH conditions, including pH 10, without requiring any enzyme engineering or modifications—an advantage compared to some alkaliphilic laccases. While BOD enzymes have primarily been employed in biomedical applications, such as biosensors and enzymatic biofuel cells [15,16,17,18], their potential for lignin modification remains unexplored. To our knowledge, no study has yet investigated the action of BOD on lignin structure.

In this work and for the first time, we investigated the enzymatic treatment of Kraft lignin under alkaline conditions at room temperature using the wild-type BOD from *Bacillus pumilus* without the addition of a mediator. This approach is particularly interesting because lignin is not a natural substrate for BOD, and the mechanism by which this oxidase interacts with the characteristic linkages in lignin remains unknown.

## 2. Experimental Section

### 2.1. Materials

Indulin AT Kraft Lignin, a pine softwood lignin, was obtained as a brown powder from Ingevity and used without further purification.

The bacterial multicopper oxidase used for this research, Bilirubin Oxidase from *B. pumilus,* was produced and purified as previously reported [19]. The enzyme activity was determined spectrophotometrically using the ABTS method. A BOD activity ranging from 700 to 900 U/mg was used throughout this study.

Boric acid (≥99.5% purity) and NaOH (≥98% purity) were purchased from Sigma-Aldrich (St Quentin FAllavier, France). DMSO-d6 (99.8%D purity) and D_2_O (99.9%D purity) were purchased from Eurisotop (St Aubin, France). All the reactants were of analytical grade and used without further purification.

### 2.2. Methods

#### 2.2.1. BOD-Lignin Reactions

Lignin was solubilized in borate buffer pH 10 (50 mM) at a concentration of 10 g·L^−1^. The BOD (3.3 mg/g of lignin) was added to the solution, and the reaction was carried out under stirring with oxygen bubbling at 22 °C during 16 h. The reaction medium is then transferred into a dialysis membrane (Spectra/Por™ Pre-wetted Standard RC Dialysis Tubing, 10 kDa MWCO) (Spectrum, Rancho Dominguez, CA, USA) in 2 L of milliQ water at 4 °C during 16 h, with one change of the dialysate, in order to remove the borate salts. The solubilized part was separated from the insoluble part by centrifugation (4500 rpm, 5 °C, 20 min). The supernatant (soluble fraction) was freeze-dried separately from the pellet (insoluble fraction).

A control experiment was performed for each enzymatic treatment, according to the same protocol, without adding the BOD in the medium.

#### 2.2.2. Characterization of Lignin


**NMR spectroscopy**


HSQC NMR experiments were performed on an Avance NEO 400 MHz spectrometer (Billerica, MA, USA), equipped with a 5 mm BBO Cryoprobe Prodigy at 296 K using the hsqcetgpsisp2.2 pulse sequence. The number of points was set to 2048 in the ^1^H dimension, and 32 scans were collected with a recycle delay of 2.5 s. In the ^13^C dimension, 256 points were recorded with a ^1^JCH coupling constant fixed at 145 Hz. The base frequencies for proton and carbon were 400.33 MHz and 100.67 MHz, respectively. Zero filling was applied in both dimensions, resulting in 2048 points in the second dimension and 1024 points in the first dimension. Each analysis was conducted on 20 to 30 mg of lignin dissolved in 0.5 mL of deuterated solvent, typically DMSO-d6, and occasionally D_2_O. In the latter case, the residual deuterated solvent signal was used as a reference for calibration (δ^13^C: 39.5 ppm; δ^1^H: 2.50 ppm).

The CP/MAS ^13^C NMR measurements were performed at 25 °C with a Bruker Avance II 400 spectrometer (Billerica, MA, USA). The operating frequency carbon is 100.63 MHz. The conventional CP/MAS method was used for high-resolution solid-state ^13^C measurements. The zirconium oxide rotors, which contain samples, were spun at 8 kHz. Other acquisition parameters were 2.8 µs of ^1^H pulse duration (90° pulse), 5 s of recycle delays, 34.41 ms of acquisition time, 30 KHz of spectral width. ^13^C Chemical shifts were calibrated through the carbonyl carbon resonance of glycine as an external reference at 178 ppm and converted to the values from tetramethylsilane (TMS). All NMR spectra were phased and baseline corrected.

The hydroxyl group content of lignin samples was determined by quantitative ^31^P NMR following a modified procedure based on Argyropoulos’ method [1]. Precisely weighed dry lignin samples were dissolved in a pyridine/deuterated chloroform mixture (1.6:1) in the presence of a known amount of the internal standard N-Hydroxy-5-norbornene-2,3-dicarboxylic acid imide. The resulting solution was derivatized using 2-chloro-4,4,5,5-tetramethyl-1,3-2-dioxaphospholane. The ^31^P NMR spectra were recorded on a 400 MHz Bruker Avance III NMR spectrometer at 298 K using a standard phosphorus pulse program. A recycling delay of 5 s was applied, and 256 scans were collected within a chemical shift range from 120 to 180 ppm. Data processing was performed using Topspin software (4.5.0), including phase and baseline correction. The spectra were calibrated using the internal standard signal at δ = 151.9 ppm.

^11^B NMR experiments were performed on a 400 MHz Bruker Avance III NMR spectrometer at 298K using the “zgbs” pulse sequence. A total of 1024 scans were acquired with a spectral width of 201.7 ppm, corresponding to an acquisition time of approximately 4 min. The sample was dissolved in DMSO-d6 and placed in a 5 mm quartz NMR tube (Norell S-5-500-QTZ-7).


**Size-exclusion chromatography**


Lignin molar masses were determined by Size-Exclusion Chromatography (SEC) using water at a pH of 12 (adjusted by NaOH addition) as the eluent. Measurements were performed with an UltiMate 3000 system from Thermo Scientific (Waltham, MA, USA) equipped with a diode array detector. The system also includes a multi-angle light-scattering detector and a differential refractive index detector from Wyatt technology (Santa Barbara, CA, USA). Polymers were separated on Tosoh PW (7.5 × 300) columns (exclusion limits from 100 Da to 300,000 Da) at a flowrate of 1 mL/min. The Columns’ temperature was held at 25 °C. Polystyrene sulfonates (PSS) were used as standards, with a mass range between 891 and 258,000 g/mol. Powders were solubilized in NaOH solution at pH 12 over 24 h, and ethylene glycol was used as a flow marker. Samples were filtrated through Nylon-66 microfilters (0.2 µm pore size) prior to injection into columns.


**Thermogravimetric Analysis (TGA)**


Thermogravimetric analyses were performed using a TGA-Q500 apparatus manufactured by TA Instruments (New Castle, DE, USA). Lignin samples weighing between 5 and 15 mg were used. Heating ramps were set at 10 °C/min under a N_2_ atmosphere from 20 °C to 800 °C, followed by heating under air up to 900 °C.


**Dynamic Light Scattering (DLS)**


Z-average hydrodynamic diameters (Dz) and the polydispersity index of the nanoparticles were determined by DLS with a Zetasizer Nano ZS from Malvern Instruments (Malvern, UK) operating with a He–Ne laser source (wavelength 633 nm, scattering angle 90°). The correlation functions were analyzed using the cumulant method. The dispersions were analyzed at 0.1 wt% in water. A backscatter angle of 173° was performed on each sample at 25 °C, followed by a 120 s equilibration period. The standard operating procedure used the refractive index (1.59) and absorption (0.01) of polystyrene latex for particle size determination. Three analyses were performed on each sample, with 10–20 runs of measurements for each repetition.

## 3. Results and Discussion

Since bilirubin oxidases (BODs) require oxygen as the final electron acceptor, all solutions were pre-saturated with oxygen at room temperature prior to use in lignin modification. Enzymatic treatments were conducted in borate buffer at pH 10 (the buffer used for the production of the enzyme), which offers a favorable environment, as most of the phenolic and carboxylic groups are deprotonated under alkaline conditions, with pKa values ranging from 7.4 to 11.4 [20]. This ensures efficient solubilization of lignin substrates. The temperature was set at 22 °C, and 10 kDa membranes were used for dialysis steps. To gain further insights into the system, control solubilizations of Kraft lignin in borate buffer at pH 10 were carried out following the same protocol, but without enzyme addition.

### 3.1. Control Experiments: Boron Effect on Lignin Structure

#### 3.1.1. Control Solubilization Kinetics

Control solubilizations were performed for 1 h, 24 h, and 48 h, followed by dialysis to eliminate the salts and centrifugation. This centrifugation step generates two fractions (the supernatant: soluble fraction and the pellets: insoluble fraction) that are separately dried by the freeze-drying technique. As results are reported in Figure 1A and Appendix A, the Kraft lignin as well as the fractions obtained from this control were analyzed by SEC in sodium hydroxide at pH 12.

According to PSS calibration, the initial molar mass of Kraft lignin was 2200 g/mol. For the insoluble fractions, a significant shift in the SEC profiles toward higher molar masses was observed, corresponding to a 3.2-fold increase in molar mass after 48 h. This increase likely reflects a fractionation phenomenon: the buffer gradually solubilized more lignin over time, leaving only the less soluble, higher molar mass chains at the end of the solubilization process. Molar masses of soluble fractions increased modestly with a 1.7-fold increase after 48 h. The molar mass increase in both fractions from the control solubilization prompted us to conduct additional analyses after the overnight reaction (16 h).

#### 3.1.2. Fractions from the Control Solubilization Analysis

Thermogravimetric analysis (TGA) under nitrogen (20–800 °C) followed by air (800–900 °C) shows similar degradation of samples around 324 °C (Appendix A). However, a significant difference in residue content is observed after pyrolysis: Kraft lignin and the insoluble fraction leave minimal residue (0.9% and 2.1%, respectively), while the soluble fraction shows 18% inorganic mass content. This unusually high value suggests the presence of salts, possibly boron salts, despite two successive dialysis baths.

Boron NMR was performed to investigate this hypothesis (Figure 2). To this purpose, quartz tubes were used and the NMR method was adapted according to Macho et al. by using the “zgbs” pulse program [21]. Samples were solubilized in DMSO-d6. The boric acid (B(OH)_3_) spectrum (a) shows a major signal at 20 ppm, broadened due to fast quadrupolar relaxation of the boron nucleus in a non-spherical electronic environment. Upon adding NaOD, shifting to alkaline conditions, sodium tetrahydroxyborate (NaB(OH)_4_) (b) appears at 1.31 ppm, with a narrow signal due to its spherical symmetry. ^11^B NMR spectra of the Kraft lignin (c) and insoluble fraction (e) show no signal. The soluble fraction (d) presents a broad signal at −3.90 ppm, indicating a new boron species distinct from (B(OH)_3_) and (NaB(OH)_4_). The high-fielded chemical shift in spectrum (d) suggests a tetracoordinated boron moiety. This result is consistent with thermogravimetric analysis detecting inorganic residues.

Boron–lignin interactions have been explored in various contexts within the literature, emphasizing the chemical reactivity of boron-containing compounds towards lignin hydroxyl functionalities. It is well established that boronic acids bind with diol-containing compounds through reversible boronate ester formation with high affinity [22]. Korich et al. demonstrated the formation of arylboronate ester linkages between the 1,3-diols of a synthetic lignin model and arylboronic acids, showcasing the feasibility of such modifications even under mild conditions [23]. In a subsequent study, the authors prepared graft copolymers by covalently linking boron-terminated polycaprolactone to organosolv lignin via reversible arylboronate ester bonds, confirmed through a combination of ^1^H, ^13^C, and ^11^B NMR analyses [24]. More recently, Dong et al. investigated the role of ammonium borate as an additive during lignin pyrolysis [25]. They found that boron incorporation increased lignin’s glass transition temperature, thermal stability, and specific surface area while altering pyrolysis product distributions. Notably, higher concentrations of simple phenols, including diphenols, were observed, suggesting that boron plays a critical role in tuning lignin’s thermal degradation pathways. Considering experimental evidence, we assume that lignin aliphatic alcohol groups are complexed by borate species, leading to the formation of B(OR)_4_^−^ moieties, as shown in Figure 3.

This result is supported by quantitative ^31^P NMR analysis (Figure 4A and Appendix A), which quantifies the hydroxyl groups in the fractions obtained from the control and compares them with the hydroxyl content of the initial Kraft lignin. Only the soluble fraction of the control was analyzable by quantitative ^31^P NMR after derivatization. A significant decrease in aliphatic hydroxyl content was observed, dropping from 1.96 to 1.48 mmol/g. Moreover, subtle reductions were noted for phenolic hydroxyls (from 3.17 to 2.83 mmol/g) and carboxylic acids (from 0.39 to 0.35 mmol/g). The observed decrease of 25% in aliphatic hydroxyl groups supports the hypothesis of borate species complexation with lignin hydroxyl groups. Additionally, another control was conducted in a phosphate buffer at pH 10 (50 mM). Comparable proportions of soluble and insoluble lignin fractions were obtained after dialysis and centrifugation. However, no variation in molar masses was observed by SEC in NaOH at pH 12, and quantitative ^31^P NMR showed consistent quantification between the initial Kraft lignin and the soluble fraction obtained from the control solubilization. This result further reinforces the role of boron in modifying the lignin structure.

### 3.2. BOD-Induced Kraft Lignin Polymerization Under Alkaline Aqueous Conditions

#### 3.2.1. BOD Treatment Kinetics

BOD enzymatic treatments were performed for 1 h, 24 h, 48 h, and 5 days, followed by dialysis, centrifugation to separate the supernatants (soluble fractions) and the pellets (insoluble fractions), and then freeze-drying. The obtained fractions were analyzed by SEC in sodium hydroxide solution at pH 12 (Figure 1B and Appendix A).

Upon enzymatic treatments with bilirubin oxidase (BOD), significant shifts in the SEC profiles toward higher molar masses are observed in both soluble and insoluble fractions. The molar mass of insoluble fractions increased up to 4-fold after 24 h of incubation, suggesting rapid enzymatic coupling reactions. However, insoluble fractions resulting from treatments longer than 24 h (48 h and 120 h) could not be solubilized in pH 12 sodium hydroxide solution, preventing their analysis by SEC. This insolubility may imply extensive polymerization or structural changes leading to a strong increase in the molar masses.

For the soluble fractions, a more progressive increase in molar mass is observed over time, reaching a 3-fold increase after 5 days (120 h) of incubation. This meaningful molar mass increase in soluble fractions may indicate oxidative coupling facilitated by BOD, although at a slower rate compared to the insoluble fractions. These findings highlight the behavior of BOD on Kraft lignin: rapid polymerization leading to insoluble fractions at shorter times and continued coupling activity in the soluble fractions over prolonged incubations. The structural changes associated with these observations were further analyzed to elucidate the enzymatic treatment mechanisms.

The enzymatic polymerization of Kraft lignin via bilirubin oxidase (BOD) enzymatic treatment is consistent with observations reported in the literature for lignins modified by alkaliphilic laccases. Several studies highlight significant molar mass increases upon enzymatic treatment under alkaline conditions, but none of them discussed the ratio between soluble and insoluble fractions of lignin after enzymatic treatment. For example, Moya et al. showed that the bacterial laccase from *Streptomyces ipomoea* induced more and more Mw increase for softwood and hardwood Kraft lignins when the pH increases from 7 to 9 [26]. Weihua et al. used *Mycelia sterilia* laccase to polymerize alkali lignin at pH 10, achieving increased Mw and decreased polydispersity [27]. More recently, Wang et al. used a bacterial laccase from MetZyme at pH 10 to polymerize fractionated alkali lignin, achieving a 13.1-fold Mw increase in just six hours, highlighting the efficiency of phenol-rich, low-Mw lignin [28]. Lastly, Lu et al. explored a co-polymerization approach between lignin and tannin-rich extracts, catalyzed by MetZyme laccase at pH 10.5, resulting in covalent bond formation (C–C and C–O–C) [29]. Similarly, in the presence of *Bacillus ligniniphilus L1* instead of isolated enzymes, Morales et al. reported a 6.75-fold Mw increase after alkaline lignin biodegradation compared to acidic conditions, demonstrating a clear polymerization trend under alkaline conditions [29].

#### 3.2.2. Fractions from the BOD Treatment Analysis

The soluble and insoluble fractions resulting from the enzymatic treatment of Kraft lignin with BOD in borate buffer at pH 10 for 16 h were selected for further detailed analysis. They are referred to as “Sol_BOD” and “Ins_BOD”, respectively.

Hydroxyl moieties were quantified via quantitative ^31^P NMR (Figure 4). The insoluble fraction “Ins_BOD” could not be analyzed using this method because the derivatization mixture precipitated in the NMR tube.

A decrease in hydroxyl content was observed in the soluble fraction. Aliphatic and phenolic hydroxyls showed a 20% reduction, while acids underwent a 30% decrease. The 20% decrease in phenolic hydroxyl groups, coupled with the significant increase in molar mass, may suggest an aryl-ether Ar-O-Ar coupling, corresponding to a 4-O-5′ linkage.

This hypothesis is supported by the solid-state ^13^C NMR spectrum (Figure 5), which shows an increased C_Ar_O signal correlated with a decline in the C_Ar_H signal. The reduction in phenolic content in lignins by enzymatic treatment has been demonstrated in various studies. Wang et al. showed that Metzyme^®^ reduced phenolic content by 30% to 70% in birch and spruce alkali lignins [28], while Mayr et al. reported a 30% to 65% reduction in softwood and hardwood Kraft lignins using CotA laccase [30]. Prasetyo et al. achieved 39% to 47% reductions in lignosulfonates with *Trametes villosa* and *Trametes hirsuta* [31], and Gouveia et al. showed that MtL laccase lowered eucalypt Kraft lignin phenolics by 66% [32]. The solid-state ^13^C NMR spectrum reveals an increased C_Ar_C signal and a decrease in the C_Ar_H signal. This fact, also linked to the substantial molar mass increase, indicates a biphenyl coupling corresponding to a 5-5′ linkage.

The 20% reduction in aliphatic hydroxyl groups observed in the phosphorus NMR analysis (Figure 4) is likely due to the presence of boron in the buffer. As demonstrated earlier, borate species can complex with the aliphatic hydroxyl groups of lignin in solution, thereby reducing the quantity of free aliphatic OH groups. This interaction also leads to a slight increase in the molecular weight of the lignin.

These findings led us to conclude that the proposed structure for the BOD-induced polymerized lignin involves 5-5′ and 4-O-5′ couplings, combined with the formation of tetracoordinated boron complexes in the borate buffer at pH 10, as shown in Figure 6.

Unfortunately, no satisfactory signal was obtained in the 2D-HSQC NMR characterization of the soluble fraction. An initial attempt was performed in DMSO-d6, and despite increasing the number of scans to 256 and optimizing D1 to 2.5 s, an extremely noisy and unusable signal was recorded. Seeking to exploit the water-solubility of the enzymatically treated soluble fraction, we conducted the acquisition in D_2_O (Appendix A). However, once again, a highly noisy signal was observed, notably with no correlation spots in the aromatic region, leading us to put this analysis aside.

A possible explanation for the absence of correlation signals in the aromatic region could be that the aromatic protons have been consumed and are therefore no longer detectable. This interpretation is consistent with our findings, which indicate that the Kraft lignin underwent enzymatic polymerization/condensation, particularly through 5–5′ coupling, leading to a decrease in the number of aromatic protons in the resulting product. This evidence matches the observations of Ibarra et al., who reported a similar disappearance of aromatic proton signals in HSQC spectra following laccase treatment [37]. They attributed this phenomenon to substantial modifications of the lignin aromatic backbone. Nevertheless, their ^13^C NMR and FTIR analyses demonstrated that the aromatic skeleton remained intact, suggesting that the observed loss of proton signals resulted from deprotonation of benzenic rings due to polymerization or condensation reactions, including the formation of α–5′ condensed structures. However, it is unexpected that all aromatic protons would be completely consumed, resulting in a total absence of signal.

Another hypothesis that could explain the inability to obtain a signal, despite the apparent clarity of the solutions in the deuterated solvents, is the presence of suspended objects invisible to the naked eye. To test this hypothesis, DLS analyses were performed using water as the solvent, adjusted to pH 12 or pH 7 depending on the solubility of the lignin samples. Kraft lignin is only soluble under alkaline conditions. The lignin samples were diluted to 1 mg/mL and appeared visually clear. The resulting DLS profiles are presented in Figure 7.

In DLS analysis, the count rate is a crucial parameter that reflects the intensity of photons scattered by particles in solution. It represents the number of photons detected per unit of time and is influenced by both the concentration and size of the particles. The count rate is used to assess the stability and quality of measurements: a stable and low variation in count rate suggests sample stability, whereas sudden or significant fluctuations may indicate phenomena such as particle aggregation or changes in concentration. In this study, the count rate values provide insights into the presence of scattering particles in the medium. Detailed numerical data from the DLS analyses are presented in Table 1.

Considering the intensity curves, all show the presence of particles with a mean size of roughly 200 nm. Nevertheless, the number-based curve, represented by dashed lines, indicates that smaller particle populations are predominant. The few larger particles scatter light more effectively since scattering intensity is proportional to particle size. Notably, the soluble fraction at pH 12 exhibits a higher count rate compared to Kraft lignin at the same pH, suggesting a greater number of scattering objects in the Sol_BOD fraction. It is also interesting to compare the count rates of the Sol_BOD fraction at the two different pH values. DLS results reveal significant differences between the Sol_BOD fractions at pH 7 and pH 12. The Sol_BOD fraction at pH 7 exhibits a very high count rate of 13,740 kcps, indicating much stronger light scattering compared to the pH 12 fraction, which has a count rate of only 1226 kcps. This difference suggests that the pH 7 sample contains more particles or that its particles scatter light more efficiently. These observations highlight the strong influence of pH on the size and distribution of lignin particles. Overall, it has been shown that the enzymatically polymerized soluble fraction, Sol_BOD, still contains scattering particles that can be considered as insolubilized material. These particles potentially remain from the preparation process but can hinder proper 2D-HSQC NMR characterization of the solubilized molecules.

## 4. Conclusions

This study demonstrates that bilirubin oxidase (BOD) effectively catalyzes the polymerization of Kraft lignin under alkaline aqueous conditions, resulting in significant increases in molar mass and structural transformations. The use of borate buffer at pH 10 not only provides optimal solubilization but also introduces boron–lignin interactions that further influence lignin structure. Control experiments highlighted the role of borate species in complexing lignin hydroxyl groups, which were supported by NMR and thermogravimetric analyses. Enzymatic treatment led to oxidative coupling reactions, as evidenced by reduced hydroxyl contents and the formation of 4-O-5′ and 5-5′ linkages, confirmed by ^31^P and ^13^C NMR. Challenges in HSQC NMR characterization suggested polymerization-induced aggregation, corroborated by DLS measurements. Beyond these structural modifications, polymerized lignin could find applications in carbon fibers synthesis, emulsion stabilization, adhesive formulations, and adsorption of pollutants. The soluble polymerized fraction could be a good candidate to spin carbon fibers [38]. Its amphiphilic properties may facilitate Pickering emulsion formation or encapsulation [39,40], while its enhanced reactivity and crosslinking potential could improve bio-based adhesives [41]. Additionally, its high molar mass and porosity could make it an effective adsorbent for wastewater treatment [42]. Collectively, these results provide mechanistic insights into enzymatic lignin modification and highlight bilirubin oxidase as a promising catalyst for lignin valorization.

## Figures and Tables

**Figure 1 polymers-17-00779-f001:**
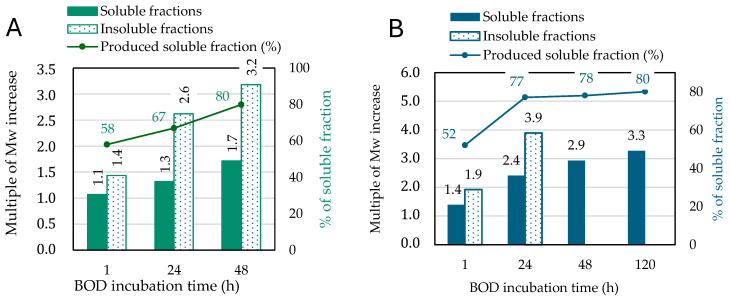
Evolution of the molar mass compared to the initial molar mass of Kraft Lignin (equal to the value 1) at different reaction times: (**A**) controlled solubilizations without enzyme; (**B**) after BOD enzymatic treatment.

**Figure 2 polymers-17-00779-f002:**
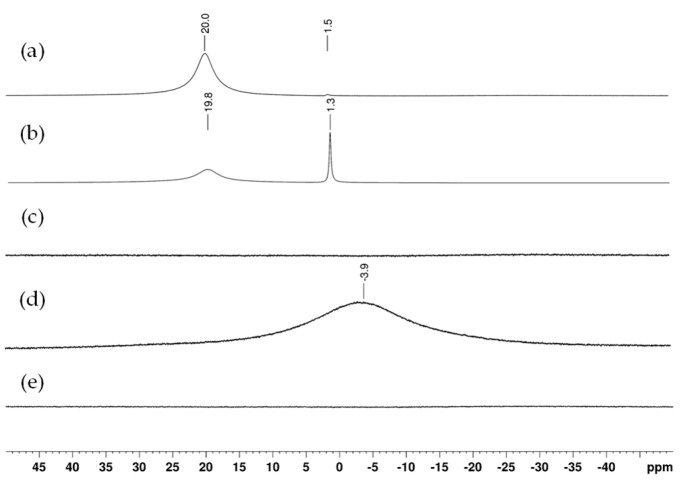
^11^B NMR at room temperature spectra in DMSO-d6, at room temperature of (a) boric acid; (b) sodium tetrahydroxyborate; (c) Kraft lignin; (d) soluble fraction from the control solubilization; (e) insoluble fraction from the control solubilization.

**Figure 3 polymers-17-00779-f003:**
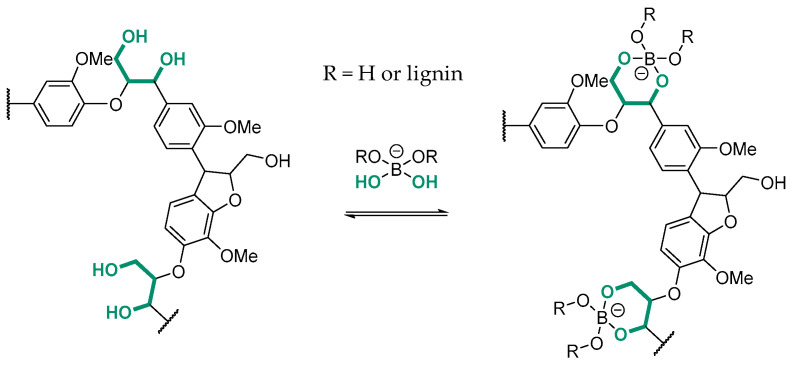
Proposed structure of the tetracoordinated boron complexes formed during lignin solubilization in borate buffer at pH 10.

**Figure 4 polymers-17-00779-f004:**
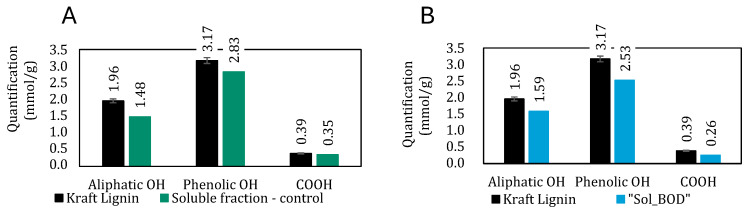
Content of different hydroxyl groups in Kraft lignin and in “Sol_BOD” determined by ^31^P NMR analysis. Error bars represent ± s.d. of five individual measurements: (**A**) controlled solubilizations without enzyme; (**B**) after BOD enzymatic treatment.

**Figure 5 polymers-17-00779-f005:**
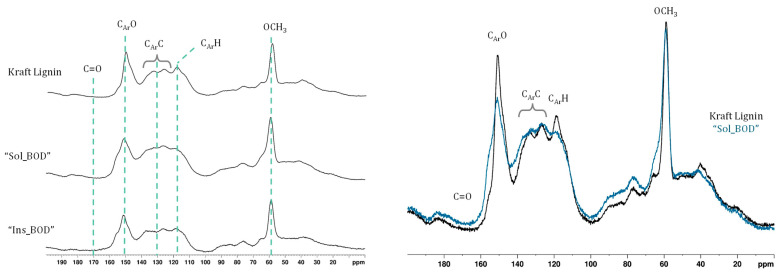
Solid-state ^13^C NMR spectra of the starting Kraft lignin and the fractions from the BOD enzymatic treatment “Sol_BOD” and “Ins_BOD”. The assignment of chemical shifts was performed using data from the literature [33,34,35,36].

**Figure 6 polymers-17-00779-f006:**
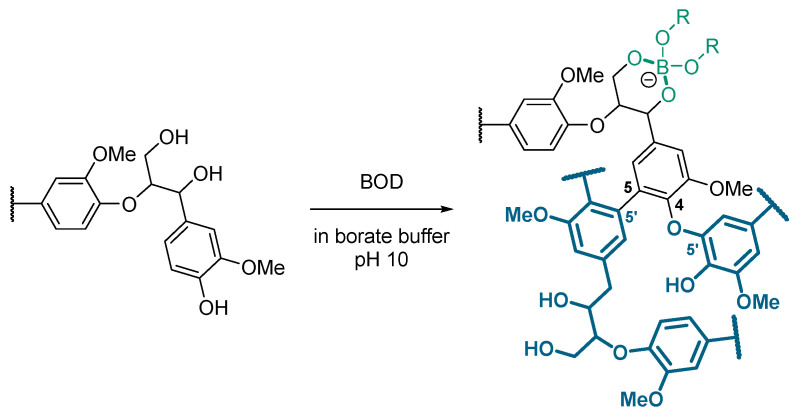
Proposed structure for the BOD-induced polymerized lignin: 5-5′ and 4-O-5′ coupling, paired with the tetracoordinated boron complexes formation in the borate buffer pH 10.

**Figure 7 polymers-17-00779-f007:**
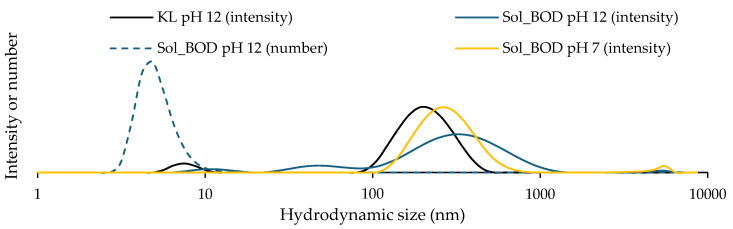
Dynamic Light Scattering (DLS) analysis of Kraft lignin and soluble fractions after enzymatic treatment with BOD: size distribution profiles at pH 7 and 12 (number- and intensity-weighted, mean diameters in nm).

**Table 1 polymers-17-00779-t001:** Dynamic light scattering measurements of kraft lignin and soluble fractions after enzymatic treatment with BOD: count rate, hydrodynamic diameter, and polydispersity index (PDI).

Sample	Count Rate (kcps)	Hydrodynamic Diameter (nm)	PDI
Kraft lignin pH 12	373	188	0.29
Sol_BOD pH 12	1226	61	0.61
Sol_BOD pH 7	13,740	181	0.31

## Data Availability

This article has all the data that were collected or analyzed during this study.

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
