# Peer review of "Homogeneous Polymerization of Kraft Lignin Using an Alkaliphilic Multi-Copper Oxidase (Bilirubin Oxidase) in a Borate Buffer"

_polymers, 2025, doi:10.3390/polym17060779_

Round 1
Reviewer 1 Report
Comments and Suggestions for Authors
The manuscript entitled “Homogeneous Polymerization of Kraft Lignin Using an Alkaliphilic Multi-Copper Oxidase (Bilirubin Oxidase) in a Borate Buffer” investigates the enzymatic modification of Kraft lignin under alkaline conditions using bilirubin oxidase (BOD) in borate buffer (pH 10). In general, the experimental is well designed, and the conclusion is well supported by the findings. Therefore, this reviewer recommends the paper for publication after addressing the following comments.
- The reaction is carried out at room temperature, but the room temperature may fluctuate and affect the reproducibility of the reaction. It is recommended to use a thermostatic device (e.g., water bath or thermostatic shaker) to control the reaction temperature to ensure consistent conditions for each experiment. The reaction time “overnight” is vague, it is recommended to specify the reaction time (e.g. 16 hours or 24 hours) to better control the reaction process.
- The mechanism of BOD-catalyzed lignin polymerization mentioned in the article is not fully understood, especially how BOD interacts with the characteristic bonds of lignin. More sentences should be added to discuss the interaction mechanism between BOD and lignin.
- The meaning of abbreviations such as “1h sol” is not mentioned in the whole paper. 1h sol and 1h ins in FigureS1 and FigureS3 are very similar in color and very light, which is difficult to see clearly.
- The article mainly focuses on the polymerization process and structural changes of lignin, but the specific applications of the polymerization products are less explored. It is suggested that the potential applications of these polymerization products in fields such as materials science, energy storage or biomedicine should be further prospected in the conclusion section in order to promote the high-value utilization of lignin.
- More relevant literature should be cited in the discussion section to more comprehensively compare the advantages and disadvantages of BOD treatment with other lignin modification methods (e.g., chemical polymerization, physical treatment) in order to highlight the uniqueness and advantages of BOD treatment.
Author Response
comment 1: The reaction is carried out at room temperature, but the room temperature may fluctuate and affect the reproducibility of the reaction. It is recommended to use a thermostatic device (e.g., water bath or thermostatic shaker) to control the reaction temperature to ensure consistent conditions for each experiment. The reaction time “overnight” is vague, it is recommended to specify the reaction time (e.g. 16 hours or 24 hours) to better control the reaction process.
response 1: The reaction time and the temperature were added in experimental part.
comment 2: The mechanism of BOD-catalyzed lignin polymerization mentioned in the article is not fully understood, especially how BOD interacts with the characteristic bonds of lignin. More sentences should be added to discuss the interaction mechanism between BOD and lignin.
response 2: We are not entirely sure what the question refers to. Are you asking about physical or chemical interactions between the enzyme and lignin? If so, we do not have a clear understanding of the mechanism. It is possible that BODs behave similarly to laccases (MCOs), but we have no certainty on this. Ongoing work on small molecules is currently in progress to further investigate this aspect.
Comment 3: The meaning of abbreviations such as “1h sol” is not mentioned in the whole paper. 1h sol and 1h ins in FigureS1 and FigureS3 are very similar in color and very light, which is difficult to see clearly.
Response 2: The abbreviations “1h sol” has been defined in Figure S1 legend.
Comment 4: The article mainly focuses on the polymerization process and structural changes of lignin, but the specific applications of the polymerization products are less explored. It is suggested that the potential applications of these polymerization products in fields such as materials science, energy storage or biomedicine should be further prospected in the conclusion section in order to promote the high-value utilization of lignin.
Response 4 :We have added a paragraph (L. 430-437) to clarify this point.
“Beyond these structural modifications, polymerized lignin could find applications in carbon fibers synthesis, emulsion stabilization, adhesive formulations, and adsorption of pollutants. The soluble polymerized fraction could be a good candidate to spin carbon fibers. [39] Its amphiphilic properties may facilitate Pickering emulsion formation or encapsulation [40, 41], while its enhanced reactivity and crosslinking potential could improve bio-based adhesives [42]. Additionally, its high molar mass and porosity could make it an effective adsorbent for wastewater treatment [43]”
Comment 5: More relevant literature should be cited in the discussion section to more comprehensively compare the advantages and disadvantages of BOD treatment with other lignin modification methods (e.g., chemical polymerization, physical treatment) in order to highlight the uniqueness and advantages of BOD treatment.
Response 5: The main advantage of this treatment is that it operates on the entire Kraft lignin without requiring any pre-treatment or fractionation. In our case, with enzymatic treatment using BOD, the majority of the starting material (80%) is recovered as water-soluble in neutral conditions (Figure 1). We have added a sentence in the introduction (lines 50-55) to clarify this point.
“Enzymatic lignin modification has gained significant interest due to its ability to operate under mild aqueous conditions, avoiding the use of high pressure, extreme temperatures, or hazardous chemical reagents. This approach enables selective oxidation and depolymerization while preserving key functional groups, making it a more sustainable and controlled alternative to conventional chemical or physical treatments.”
Reviewer 2 Report
Comments and Suggestions for Authors
- Abstract is well constructed, I have a technical question here, the study reports a molar mass increase of up to 7-fold in the control solubilization without enzyme addition, potentially due to borate complexation with lignin hydroxyl groups. How do the thermogravimetric and NMR analyses specifically support this hypothesis.
- Introduction section contains so much literature relevant to the presented study but it is difficult to find that authors have also reported the outcomes of those studies, please mention the outcome relevant to presented study so that readers can compare the results from the literature.
- There are several typo and grammatical mistakes in manuscript;
Line 34, today. [3, 4].
Introduction, line 62, “as substrates However, the”
Line 64, double full stop “can degrade car-63 bonization.. Other”
Line 199, correct the sentence “Results are gathered on Figure 1A and Figure S1.” As results are reported in……
Etc.
- Authors are advised to mention the chemical grading and the country for the chemicals they have used. “Boric acid, NaOH, DMSO-d6, D2O were purchased from Sigma-Aldrich.”
- Figure 1 is presented in descriptive way without providing any logical physical significance, like how does the observed 3.2-fold increase in molar mass of the insoluble fractions after 48 hours correlate with potential fractionation mechanisms?
- Moreover, as the initial molar mass of Kraft lignin was 2,200 g/mol, how do the reported increases in molar mass (both 3.2-fold for insoluble and 1.7-fold for soluble fractions) align with the solubilization dynamics over time. Please explain
- Line 396, “Considering the intensity curves, all show the presence of particles with a mean size of roughly 200 nm.” Please provide the elemental distribution mapping or the micrographs.
Comments on the Quality of English Language
English should be revised.
Author Response
Comment 1: Abstract is well constructed, I have a technical question here, the study reports a molar mass increase of up to 7-fold in the control solubilization without enzyme addition, potentially due to borate complexation with lignin hydroxyl groups. How do the thermogravimetric and NMR analyses specifically support this hypothesis.
Response 1: The correct value is a 1.7-fold increase, not 7. The answer to this question can be found in lines 236-240. Additionally, we have specified '¹¹B NMR analyses' in the abstract to provide greater clarity
Comment 2: Introduction section contains so much literature relevant to the presented study but it is difficult to find that authors have also reported the outcomes of those studies, please mention the outcome relevant to presented study so that readers can compare the results from the literature.
Response 2: The outcomes are detailed in the text (lines 39-50). Regarding alkaliphilic enzymes, we stated: 'These studies revealed an ambivalence in the action of these enzymes, with observations of both polymerization and depolymerization, and sometimes even simultaneous occurrences of these phenomena' (L. 78-80). We chose not to elaborate further on this aspect in the introduction but instead refer readers to the detailed review article. The key outcomes of the most relevant studies for our work are discussed in the results section, where they are compared to our findings (e.g., L. 244-258).
Comment 3: There are several typo and grammatical mistakes in manuscript;
Line 34, today. [3, 4].Introduction, line 62, “as substrates However, the”
Line 64, double full stop “can degrade car-63 bonization.. Other”
Line 199, correct the sentence “Results are gathered on Figure 1A and Figure S1.” As results are reported in……
Response 3: The typos have been corrected
Comment 4: Authors are advised to mention the chemical grading and the country for the chemicals they have used. “Boric acid, NaOH, DMSO-d6, D2O were purchased from Sigma-Aldrich.”
Response 4 : The grade of chemicals have been added
Comment 5: Figure 1 is presented in descriptive way without providing any logical physical significance, like how does the observed 3.2-fold increase in molar mass of the insoluble fractions after 48 hours correlate with potential fractionation mechanisms?
Response 5: The fractionation process occurs progressively over time. Higher molecular weight fractions remain insoluble in the borate buffer at pH 10, even after extended solubilization periods, suggesting limited solubility of those highly polymerized lignin species in this medium. In contrast, lower molecular weight and less polydisperse populations dissolve more readily in the borate buffer, indicating a preferential solubilization of smaller lignin fragments over time. By the end of the solubilization step, the insoluble fraction accounts for only 20% of the initial lignin. This observation is not fully captured by normalization of the chromatograms, which does not reflect the fractionation phenomenon (Figure S1). This increase likely reflects a fractionation phenomenon: the buffer gradually solubilized more lignin over time, leaving only the less soluble, higher molar mass chains at the end of the solubilization process (L. 215-217)
Comment 6: Moreover, as the initial molar mass of Kraft lignin was 2,200 g/mol, how do the reported increases in molar mass (both 3.2-fold for insoluble and 1.7-fold for soluble fractions) align with the solubilization dynamics over time. Please explain.
Response 6 The increases in molar mass observed (3.2-fold for the insoluble fraction and 1.7-fold for the soluble fraction) are consistent with the solubilization dynamics over time. The incorporation of anionic borate species and borate bridging interactions likely charge the lignin chains, enhancing their solubility. These interactions facilitate the solubilization of lower molecular weight fractions, while higher molecular weight fractions remain insoluble, contributing to the observed changes in molar mass.
Comment 7: Line 396, “Considering the intensity curves, all show the presence of particles with a mean size of roughly 200 nm.” Please provide the elemental distribution mapping or the micrographs.
Response 7: We are not entirely sure we understood the question. Are microscopy images being requested? We did not perform microscopy as this technique does not provide information on lignin in solution, which is the focus of our study. However, we have provided other characterization techniques that effectively analyze the solubilized lignin.
